# C3d-Positive Preformed DSAs Tend to Persist and Result in a Higher Risk of AMR after Kidney Transplants

**DOI:** 10.3390/jcm9020375

**Published:** 2020-01-30

**Authors:** Sooin Choi, Kyo Won Lee, Jae Berm Park, Kyunga Kim, Hye-Ryeon Jang, Wooseong Huh, Eun Suk Kang

**Affiliations:** 1Department of Laboratory Medicine, Soonchunhyang University Hospital Cheonan, Soonchunhyang University College of Medicine, Cheonan 31151, Korea; sooin2@schmc.ac.kr; 2Department of Surgery, Samsung Medical Center, Sungkyunkwan University School of Medicine, Seoul 06351, Korea; kw1980.lee@gmail.com (K.W.L.); jbparkmd@gmail.com (J.B.P.); 3Statistics and Data Center, Research Institute for Future Medicine, Samsung Medical Center, Seoul 06351, Korea; kyunga.j.kim@samsung.com; 4Division of Nephrology, Department of Medicine, Samsung Medical Center, Sungkyunkwan University School of Medicine, Seoul 06351, Korea; shinehr@skku.edu (H.-R.J.); whuh03@skku.edu (W.H.); 5Department of Laboratory Medicine and Genetics, Samsung Medical Center, Sungkyunkwan University School of Medicine, Seoul 06351, Korea

**Keywords:** kidney transplant (KT), donor-specific antibodies (DSA), C3d-binding assay, antibody-mediated rejection (AMR)

## Abstract

C3d-binding assays have been introduced as methods for the prediction of the presence of complement-binding functional antibodies; however, the prognostic value of C3d-positive preformed donor-specific antibodies (pDSAs) has not been fully evaluated. In this study, we performed a retrospective investigation of the association of pDSAs and their C3d-binding capacity with one-year clinical outcomes. pDSAs were defined as donor-specific antibodies (DSAs) that were produced before kidney transplants (KTs) (pre-pDSAs) or within the first four weeks after KTs, owing to rebound immune response (post-pDSAs). Of 455 adult KT recipients, pre-pDSAs and post-pDSAs were found in 56 (12.3%) and 56 (12.3%) recipients, respectively, and C3d-positive post-pDSAs were found in 13 recipients (2.9%) in total. Approximately half of the C3d-negative pre-pDSAs (37/73, 50.7%) disappeared after transplantation; however, all C3d-positive pre-pDSAs (8/8, 100%) persisted after transplantation despite desensitization (*p* = 0.008). C3d-positive pDSAs were significantly associated with a higher incidence and risk of AMR (*p* < 0.001, OR 94.467–188.934). Identification of the C3d-binding activity of pDSAs before and early after KT is important for predicting the persistence of pDSAs and the risk of AMR induced by the presence of pDSAs.

## 1. Introduction

Donor-specific human leukocyte antigen antibodies (DSAs) are a critical factor in kidney transplantation (KT), as antibody-mediated rejection (AMR) induced by the binding of DSAs to the allograft represents a major post-transplant complication. If DSAs are detected before transplantation, desensitization procedures, such as the administration of rituximab or intravenous immunoglobulin (IVIG), or plasmapheresis, may be performed to reduce the DSA titer and lower the risk of AMR [1]. 

Single antigen bead-based antibody identification (SAB) assays are generally used as the standard for DSA monitoring. However, the clinical relevance of all the detected DSAs remains unclear because the presence of DSAs does not always correlate with complement-mediated cytotoxicity crossmatching and may not induce AMR. Recently, C1q- and C3d-binding assays were introduced as methods for predicting the presence of complement-binding functional antibodies; however, the prognostic value of these tests remains controversial [2,3,4,5,6,7,8]. In particular, the clinical significance of preformed DSAs (pDSAs) with complement-binding activities has been not fully evaluated. 

Human leukocyte antigen (HLA) antibodies at a titer below the SAB assay cut-off level or diluted across multiple beads that share target epitopes may not be appropriately detected in SAB assays. The titer of those cryptic antibodies can be elevated by immunological memory response shortly after KT [9]. Wiebe et al. suggested that false-negative pDSAs and their increased titer after transplantation, due to memory B cell activation, may create the false impression of de novo DSA (dnDSA) early post-transplantation. They suggested that no dnDSA was detected prior to 6 months when using two strict definitions: (1) all historic and current samples were DSA-negative, with an MFI cut-off of 300 and special attention to grouped epitopes; (2) no AMR in protocol biopsies at 6 months after transplantation [10]. Therefore, we thought that DSAs that were found only before KT should not be defined as pDSAs. In this study, we considered all of the DSAs produced within one month following KT as pDSAs and then investigated the production of pDSAs in pre- and post-transplantation and evaluated their effect on the occurrence of acute rejection and clinical outcome associated with their C3d-binding activity. 

## 2. Materials and Methods

### 2.1. Study Population

Of the 560 adult recipients who underwent KT between January 2013 and July 2017 at the Samsung Medical Centre, Seoul, Korea, 455 patients (279 men and 176 women) were included in this study. Multi-organ, ABO-incompatible, or combined kidney and bone marrow transplantation cases were excluded (Figure 1). Recipients who underwent desensitization owing to a high level of panel reactive antibodies (PRA) (over 50%) without DSAs were excluded. All cases were negative for complement-dependent cytotoxicity (CDC), crossmatched on the day of transplantation. DSA status was monitored pre-transplantation (within 1 month before KT) and post-transplantation (at 1 and 4 weeks after KT). The recipients were divided into four groups according to the presence of DSAs before (pre-pDSA) and after transplantation (post-pDSA): Group 1, the pDSA-negative group (recipients without pre- and post-pDSAs); Group 2, the cryptic pDSA rebound group (comprising recipients with post-pDSA only); Group 3, the pDSA reversed group (recipients with pre-pDSA only); and Group 4, the pDSA persistent group (recipients with both pre- and post-pDSAs). To determine the effect of C3d-binding capacity on clinical outcomes, the post-pDSA positive groups (Groups 2 and 4) were subdivided according to their C3d-binding capacity. A schematic of the study is shown in Figure 2. This study was approved by the Institutional Review Board of Samsung Medical Centre, Seoul, Korea (SMC-2016-07-140-003), and the requirement for the subjects’ informed consent was waived.

### 2.2. Desensitization and Immunosuppression

The desensitization protocol consisted of the administration of rituximab (Genentech Inc., San Francisco, CA, USA), treatment with intravenous immunoglobulin (IVIG) (Green Cross, Seoul, Korea), and plasmapheresis by using a COBE Spectra (Gambro BCR, Lakewood, CO, USA) before transplantation. In recipients with a pre-pDSA mean fluorescence intensity (MFI) of ≥2,500 by SAB assay, all three desensitization protocols were performed; in recipients with a low pre-pDSA MFI of <2,500 by SAB assay, only rituximab was administered. For the induction of immunosuppression, recombinant anti-thymocyte globulin (rATG) (Genzyme, Cambridge, MA, USA) was administered on Day 0; it was administered post-transplantation on Days 1 and 2. Basiliximab (Simulect, Novartis, Basel, Switzerland) was used to induce immunosuppression in recipients without pre-pDSA. Desensitization and immunosuppression protocol details have been described elsewhere [11]. 

### 2.3. Immunologic Assays

HLA-A, -B, -C, -DRB1, and -DQB1 loci HLA typing for donors and recipients was performed by using polymerase chain reaction with sequence-specific primer (PCR-SSP) (One Lambda, Canoga Park, CA, USA) or reverse sequence-specific oligonucleotide probes (rSSOP) (Immucor, Peachtree Corners, GA, USA). 

Anti-HLA antibody measurements were performed at 1 and 4 weeks after KT. Anti-HLA class I and II IgG antibodies were tested by using a Luminex bead-based detection assay. The LABScreen Mixed kit (One Lambda, Waltham, MA, USA) was used to screen for class I and II antibodies in conjunction with the HLA Fusion software v3.0 (One Lambda, Canoga Park, CA, USA). Sera that were positive in the screening test were subsequently tested for HLA antibody specificities and the presence of DSA using the LIFECODES LSA Class I and Class II SAB kit (Immucor, Stamford, CT, USA) in accordance with the manufacturer’s recommendations; the results were analyzed by using Match-It software v1.2 (Immucor, Norcross, GA, USA). All sera were subjected to SAB treatment with 50 mM dithiothreitol (DTT) for 30 min. Antibody-positive results were assigned when more than two criteria were calculated from background MFI, control MFI was calculated, and the normalization factors recommended by the manufacturer were met. 

The complement-binding capacity of DSAs in sera stored at −70 °C was measured using the LIFECODES C3d assay kit (Immucor, Stamford, CT, USA) in accordance with the manufacturer’s instructions.

### 2.4. Data Collection and Statistical Methods

Data describing patient characteristics and their clinical outcomes were obtained from medical records. Categorical variables were summarized by number and percentage (%) and compared among groups by using Fisher’s exact test or the chi-square test as appropriate. For continuous variables, the Shapiro–Wilks test was performed beforehand to examine the normality of distributions, and then summarized with mean (SD, standard deviation) or median (IQR, interquartile range) and compared among groups using one-way ANOVA or the Kruskal–Wallis test according to the normality of their distribution.

#### 2.4.1. Recipient Characteristics

Data on demography, underlying diseases, transplantation conditions, conditioning, and immunosuppression regimens were included in the analysis. 

#### 2.4.2. HLA Antibody Characterization

Preformed DSAs (pDSAs) were defined as DSAs that were produced before KT (pre-pDSAs), including Groups 3 and 4, or within the first 4 weeks after KT due to rebound immune response (post-pDSAs), such as Group 2. One of our hypotheses was that the cryptic pDSA rebound group (Group 2) would be useful for the assessment of the effects of pDSAs that were undetectable before transplantation using current antibody tests, and patients were therefore not subjected to the pre-transplant desensitization, in contrast to the sensitization in the pre-pDSA positive groups (Groups 3 and 4). Receiver operating curves (ROC) were plotted to assess SAB MFI performance in an effort to predict the C3d-binding activities of HLA antibodies. Optimal cut-offs exhibiting maximal sensitivity and specificity (Youden index) were obtained for risk assessment of the presence of DSAs. 

#### 2.4.3. Clinical Outcome

To assess clinical outcomes, graft rejection rate, rejection-free survival, and graft function were evaluated. Graft biopsy was performed on Day 14 and 1 year post-KT, or whenever there was clinical suspicion of acute rejection. Biopsy results up to 400 days after KT were included in the analysis because the protocol-mandated biopsy schedule was delayed for some patients due to hospital or patient circumstances. Acute cellular rejection (ACR) and AMR were diagnosed in accordance with the Banff Criteria 2013 [12]. The rates of ACR and AMR occurrence were compared among recipient groups using Fisher’s exact test or the chi-square test, as appropriate. The univariable logistic regression was repeatedly used for the four-group comparison (Groups 1, 2, 3, and 4) and also for two- or four- subgroup comparisons: Group 1, Group 4 C3d (-) subgroup, and Group 4 C3d (+) subgroup. Wald’s chi-square test was used for pairwise comparison with Bonferroni’s correction. Rejection-free survival rates were estimated by using the Kaplan–Meier method, and the four groups were compared via the stratified log-rank test. Graft function was evaluated using the estimated glomerular filtration rate (eGFR) at 1, 3, 6, 9, and 12 months post-KT. eGFR was calculated using the modification of diet in renal disease (MDRD) study equation. Generalized estimating equation (GEE) analyses were applied to repeated measurements of eGFR. *p* < 0.05 was considered statistically significant. 

#### 2.4.4. Statistical Software

Statistical analyses were computed by using SAS v9.4 (SAS Institute, Cary, NC, USA) and SPSS v22.0 (IBM, Armonk, NY, USA). Analyse-it v5.10 (Analyse-it Software, Leeds, UK) was used for graphical analyses.

## 3. Results

### 3.1. Recipient Characteristics

Fifty-six recipients (12.3%) had detectable DSAs prior to KT (re-pDSA; Figure 1). The patients were divided into four groups according to their pre- and post-KT DSA status: the pDSA negative group (Group 1; *n* = 380, 83.5%), the cryptic pDSA rebound group (Group 2; *n* = 19, 4.2%), a pDSA reversed group (Group 3; *n* = 19, 4.2%), and a pDSA persistent group (Group 4; *n* = 37, 8.1%). All recipients with pre-pDSA (Groups 3 and 4) underwent desensitization. The recipient characteristics for each group are summarized in Table 1. 

In pre-transplantation, seven recipients (12.5%) had C3d-positive pDSAs; however, the number of recipients having C3d-positive pDSAs increased to 13 recipients (23.2%) within the first month after KT. Five recipients exhibited persistently C3d-positive pDSAs before and after transplantation. HLA classes of total and C3d (+) DSAs in Groups 2, 3, and 4 are summarized in Table 2. Class II DSAs were assessed with limited loci, as –DR and -DQB1. In contrast with the higher frequencies of class I HLAs in pre-transplantation, class II HLAs were more frequent in post-transplantation, not only among total DSAs but also among C3d (+) DSAs. However, there were no significant differences in the distribution of post-pDSA class and SAB MFI between Groups 2 and 4 (*p* = 1.000; *p* = 0.327 in class I and *p* = 0.882 in class II, respectively; Appendix A). 

### 3.2. HLA Antibody Characteristics, Including Complement Binding Capacities

In total, 105 pDSAs derived from 75 recipients (Groups 2, 3, and 4) were identified; the median number per recipient was 1.4 (range: 1–6) (Appendix A). Of these pDSAs, 81 (77.1%) were persistent pre-pDSA, and 24 (22.9%) were rebound cryptic pDSAs that were newly produced within 4 weeks post-KT. C3d-binding capacities were observed in 9.9% (8/81) of pre-pDSA and 20.6% (14/68) of post-pDSA. Among the 73 C3d-negative pre-pDSAs derived from 49 recipients, 50.7% (37/73) became negative (pDSA reversed), 43.8% (32/73) persisted as C3d-negative, and 5.5% (4/73) became C3d-positive after transplantation. Pre-pDSA SAB MFIs were significantly higher in Group 4 than in Group 3 (*p* < 0.001). Importantly, all 8 C3d-positive pre-pDSAs identified in 7 recipients persisted after transplantation, although 25% (2/8) became C3d-negative, which was significant when compared with the C3d-negative pre-pDSAs (*p* = 0.008; Figure 3, red lines). 

To determine the SAB MFI cut-offs for HLA antibodies predicting C3d-binding activities, all 1515 HLA antibodies from 112 recipients were analyzed (Figure 4). The numbers of observed HLA antibodies to A, B, C, DR, DQB1, and DPB1 were 322, 526, 71, 370, 159, and 67, respectively. SAB MFIs of C3d-positive antibodies were significantly higher than those of C3d-negative antibodies in all loci. For class I antibodies, 15.1% (139/919) of the HLA antibodies were C3d-positive, and their median SAB MFI was 9429 (IQR: 5457–16,016), whereas that of the C3d-negative antibodies was 1988 (IQR: 1271–3190). The optimal cut-off value of the total class I loci for predicting C3d-binding activities was 7797, and the area under the curve (AUC) was 0.908. For class II antibodies, 42.6% (254/596) of the HLA antibodies were C3d-positive, and their median SAB MFI was 10,341 (IQR: 6693–14,207), whereas that of the C3d-negative antibodies was 1,711 (IQR: 1002–3401). The optimal cut-off value of total class II loci was 4460, and the AUC was 0.914 (Appendix A).

### 3.3. Rejection Episodes and Graft Function

Overall, 177 recipients (38.9%) were diagnosed with rejection episodes, including ACR and AMR, with an incidence of 168 (36.9%) and 13 (2.9%), respectively; in addition, 4 (1.7%) recipients were diagnosed with both ACR and AMR (Table 3). The incidence of ACR among the four groups was not significantly different. In contrast, the incidence of AMR among the four groups was significantly different. In both Groups 2 and 4, recipients with C3d-positive post-pDSA exhibited a significantly higher incidence of AMR (2/4, 50.0% and 3/9, 33.3%, respectively) than recipients with C3d-negative post-pDSA (1/15, 6.7% and 4/28, 14.3%, respectively). The odds ratio (OR) of AMR risk was significantly increased in the Group 2 C3d (+) subgroup and both C3d (-) and (+) subgroups in Group 4 compared with Group 1 (Table 4). Although it had marginal statistical significance (adjusted *p* = 0.0876), the C3d-positive subgroup, but not the C3d-negative subgroup in Group 2, exhibited quite different OR (OR = 0.056 and OR = 0.778, respectively) compared with Group 3.

The 1-year AMR-free survival was also significantly different among all groups (Figure 5a), and it was the lowest in the C3d-positive subgroup of Group 2, the cryptic pDSA rebound cases (Figure 5b). 

During the maximum 400 day follow-up period (median 17.5, IQR 12.0–330.0), graft failure was not observed in any recipients, and the eGFR differences observed among the groups were not significantly different at any time point (*p* = 0.575; Appendix A). 

## 4. Discussion

Both preformed DSAs present before (pre-pDSAs) and early after KT (post-pDSAs) were associated with the risk of AMR when they had C3d-binding activities. In particular, C3d-positive pre-pDSAs tended to persist after transplantation, despite the pretransplant desensitization.

### 4.1. Cryptic DSAs

pDSAs that were produced within 1 month of transplantation were considered as cryptic pDSAs with anamnestic reactions [10]. Using this study design, we compared the clinical effect of pDSAs with or without pre-transplantation desensitization. Our results indicated that the presence of rebound pDSA was primarily associated with AMR, followed by that of persistent pDSA, reversed pDSA, and negative pDSA. 

The incidence of newly produced DSAs after transplantation, which usually occurs within 1 year of KT, with a variable median time between 6 months and 4.6 years, has been reported in 13% to 30% of pDSA-negative recipients before KT [10,13,14]. Several studies have analyzed the effect of early produced DSAs, which were defined as those produced within 1 year of transplantation [15,16], but it remained unclear whether this was dnDSA or pDSA [16]. King et al. reported that DSAs that were produced sooner than 1 month after transplantation exerted more pronounced effects on recipients and allografts than the effects of those produced later [16]. Cryptic DSAs may exist below the detection limit of the current antibody tests and can be induced rapidly by anamnestic reactions after re-stimulation by the donor graft. This is in contrast with dnDSAs, which develop gradually through the primary immune reaction associated with an indirect pathway after encountering new alloantigens [9,10]. Anamnestic reactions lead to the production of C3d-positive pDSAs, and this was related to the highest incidence of AMR in this study. Therefore, further studies regarding the management of cryptic pDSAs are necessary.

### 4.2. Persistent pDSAs

pDSA persistence is known to be associated with AMR. Kimball et al. reported a higher AMR incidence in persistent pDSA groups than in negatively converted groups (43% and 3%, respectively) [17]. In addition, Marfo et al. demonstrated that recipients with persistent pDSAs experienced more acute and chronic rejection (*p* = 0.006) than recipients with reversed pDSA [18]. The independent risk factor associated with persistence of pDSA was pre-transplant MFI, as shown by Redondo-Pachon et al., and class II DSAs persisted more frequently [19]. Similarly, we found that persistent pDSAs possessed higher MFIs. Although the risk of AMR was not statistically different between the persistent pDSA group and reversed pDSA group, it tended to be higher in the persistent pDSA group. The C3d-positivity (33.3%, 3/9) of persistent pDSA resulted in an increased risk of AMR (OR = 9.000, *p* = 0.0781) compared with C3d-negative group (OR = 3.000, *p* = 0.3439). 

### 4.3. Complement-Binding Capacities and Clinical Outcome

SAB was designed to detect all IgG antibody isotypes, irrespective of their complement-binding capacity. IgG3 was the most potent complement binder among the subclasses of IgG and significantly affected the occurrence of rejection and graft loss after transplantation via IgG3-induced C1q-binding [20]. Honger et al. reported that IgG pre-pDSA was composed of 39% IgG1 and/or IgG3, 7% IgG2 and/or IgG4, and a 54% mixture of both complement-binding and weak/non-complement-binding subclasses [21]. These findings suggested that not all SAB-positive antibodies promoted complement activation, rejection, and graft loss. Therefore, two modified SAB assays targeting different complement derivatives were recently introduced (the C1q- and C3d-binding assays). dnDSAs harboring C1q-binding capacity were known to affect clinical outcomes such as graft survival, acute AMR, and transplant glomerulopathy [8,22,23,24,25,26,27], but the role of pDSA in AMR or poor graft survival prediction is controversial [28,29,30,31]. C3d-positive dnDSA increased the risk of graft loss, AMR, proteinuria, C4d histological staining, and rapid progression to graft dysfunction [3,6,32,33]; however, C3d-positive pre-pDSA was reported not to increase the risk of graft failure significantly [34]. In the present study, we performed a C3d-binding assay, rather than a C1q-binding assay, for two reasons. First, targeting C3d is more relevant because C3d is derived after the initiation of the complement activation cascade, so it may reflect more of the functional aspects of antibodies [7]. Second, less data are available on the significance of C3d-positive pDSAs than that of C1q-positive pDSAs. We also found that C3d-positive pDSAs tended to persist after KT, and were associated with higher AMR incidence, regardless of desensitization. 

The threshold of SAB MFI, which was correlated with C1q- and C3d-binding capacities, has been reported [3,7], with the purpose of estimating the risk of identified antibodies in a timely manner before carrying out the subsequent complement binding assay. The suggested cut-off MFIs for predicting C3d-binding capacities in previous reports ranged from 4225–17,057 and 8356–15,027 for HLA class I and class II antibodies, respectively [6,32,35]. Class II HLA antibodies were reported to possess higher MFIs than those of class I in regard to C3d-binding capacity. In this study, the median MFI of C3d-positive antibodies also tended to be higher in class II than those of class I, but the optimal cut-off value of class I (MFI, 7797; sensitivity, 61.9%; specificity, 97.9%) was higher than that of class II (MFI, 4460; sensitivity, 88.9%; specificity, 83.9%). Such a discrepancy may be due to different sensitivities and specificities, as determined by different studies or the limited number of HLA antibodies analyzed in our study. Kamburova et al. reported that 95% of C3d-positive antibodies exhibit SAB MFI values of 4000 or more, but only 56% of antibodies exhibiting an MFI of 4000 or more possessed C3d-binding capacity. Based on this, they suggested that the C3d-binding capacity was correlated with the SAB MFI; however, positivity cannot be completely predicted based on SAB MFI [34]. 

This study has a few limitations. First, the definition of pDSAs, which included cryptic pDSAs, was not verified by using donor-specific mBCs, and the possibility of rapidly produced dnDSAs cannot be excluded. Second, the number of recipients who progressed to AMR was small, although the study was able to determine statistical significance from these data. Finally, the contribution of HLA-DP DSAs could not be estimated, as donor and recipient DP typing was not performed. 

## 5. Conclusions

The monitoring of pDSA persistence, particularly that with C3d-binding capacities that occurred despite desensitization, and of DSA production immediately following KT likely reflects that elevated cryptic pDSAs, due to the anamnestic response, are critical for the prediction of AMR. This approach would aid the initiation of timely therapeutic intervention to reduce the risk of DSAs.

## Figures and Tables

**Figure 1 jcm-09-00375-f001:**
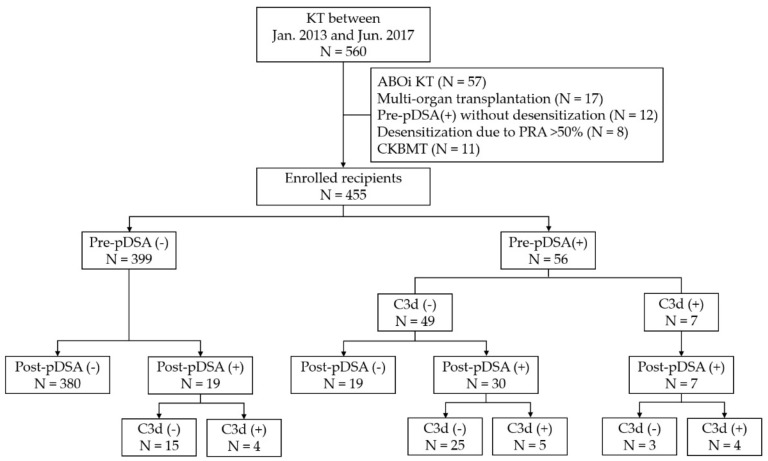
Study population and recipient groups according to donor-specific antibody (DSA) presence and C3d-binding capacity. Pre-pDSA, DSA confirmed before transplantation; Post-pDSA, DSA confirmed within 1 month of KT; KT, kidney transplantation; ABOi, ABO-incompatible; pDSA, preformed donor-specific HLA antibody; PRA, panel-reactive antibody; CKBMT, combined kidney and bone marrow transplantation.

**Figure 2 jcm-09-00375-f002:**
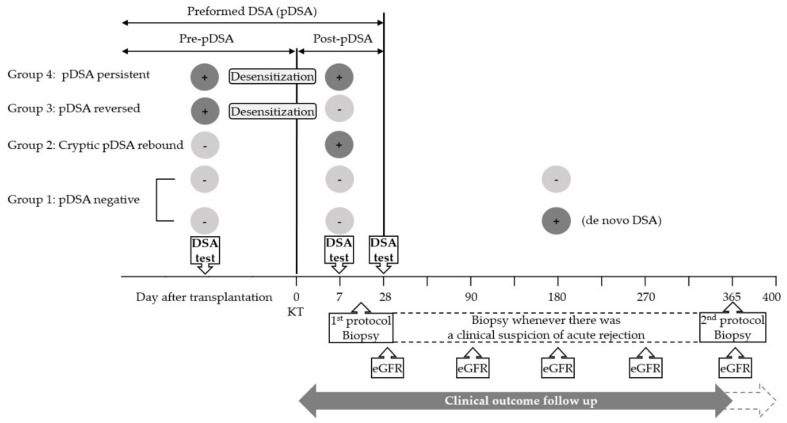
The study scheme showing definitions of recipient groups and a summary of clinical outcome follow up. All DSAs presented within 1 month before KT and produced within 1 month after KT were considered as preformed DSAs (pDSAs). Pre-pDSA, DSAs confirmed before transplantation; Post-pDSA, DSAs confirmed at 1 week and/or 4 weeks after KT; KT, kidney transplantation; pDSA, preformed donor-specific HLA antibody; SAB, single antigen bead-based antibody identification assay; eGFR, estimated glomerular filtration rate.

**Figure 3 jcm-09-00375-f003:**
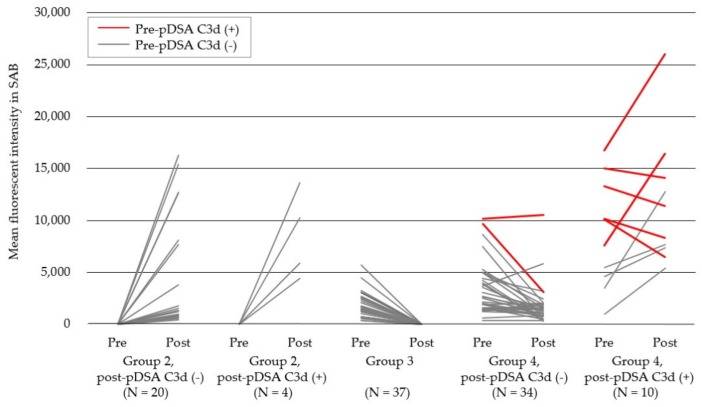
Production and MFI changes of pDSAs within 4 weeks of kidney transplantation in different patient groups. The red lines indicate cases of C3d-positive pDSAs and the grey lines highlight cases of C3d-negative pDSAs. pDSA, preformed donor-specific HLA antibody; MFI, mean fluorescence intensity.

**Figure 4 jcm-09-00375-f004:**
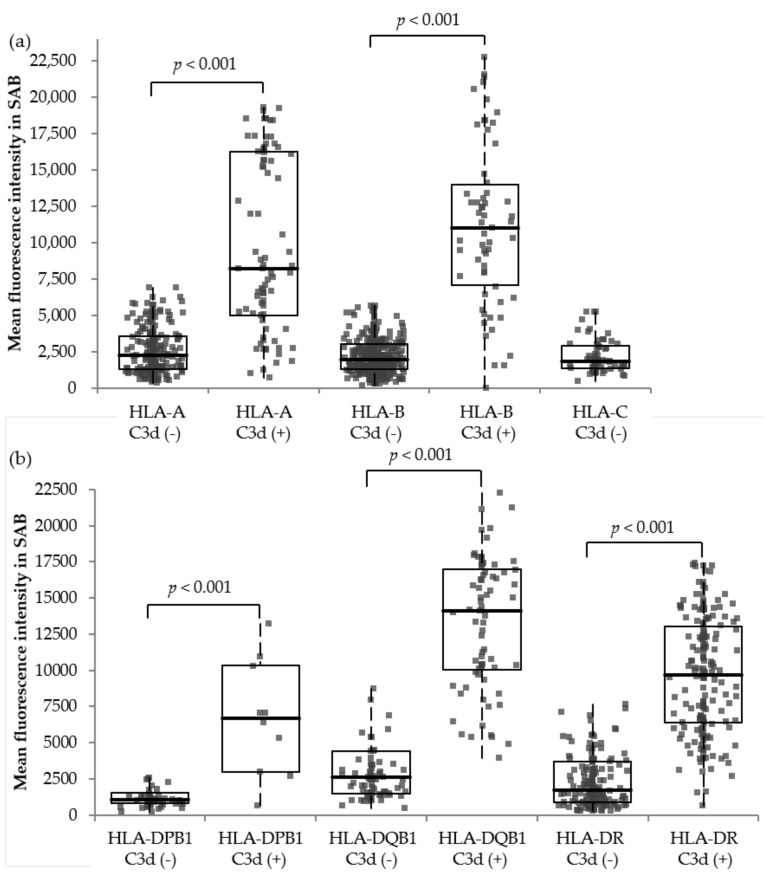
Mean fluorescence intensity distribution of 1522 Class I and II HLA antibodies according to their loci and C3d-binding capacities (**a**,**b**). Box plot, 1st to 3rd quartile range and whiskers extend to the furthest observation within 1.5× interquartile range from the quartiles. SAB, single antigen bead-based antibody identification assay.

**Figure 5 jcm-09-00375-f005:**
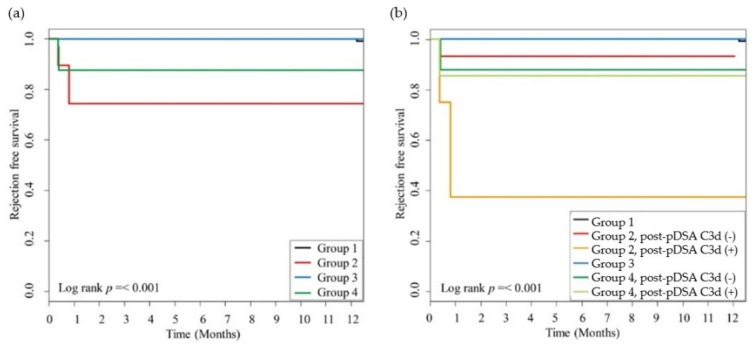
AMR-free survival according to the presence of pre-and post-pDSAs and C3d-binding capacities. The incidences of AMR were significantly different among the groups (**a**) and when considering the presence of post-pDSA C3d-binding capacities (**b**) (*p* < 0.001, respectively). The statistical difference between the Kaplan–Meier survival curves was evaluated by using the log-rank test, and *p*-values of <0.05 were considered statistically significant. AMR, antibody-mediated rejection; pDSA, preformed donor-specific HLA antibody.

**Table 1 jcm-09-00375-t001:** Patient characteristics.

Characteristics	Total	Group 1: pDSA negative	Group 2: Cryptic pDSA rebound	Group 3: pDSA reversed	Group 4: pDSA persistent	*p*-Value *
Number	455	380	19	19	37	
Age, median (IQR)	52.0 (43.0–59.0)	52.0 (42.0–59.0)	49.0 (40.0–56.3)	51.0 (44.2–57.7)	52.0 (48.0–60.0)	0.635
Sex (male) (%)	279 (61.3)	259 (68.2)	7 (36.8)	1 (5.3)	12 (32.4)	<0.001
Dialysis duration, median (IQR)	810.0 (50.2–2173.7)	843.5 (51.4–2164.1)	260.0 (47.3–2107.2)	172.0 (1.2–1652.7)	1305.0 (82.7–2428.3)	0.333
Underlying diseases (%)						
DM	125 (27.5)	114 (37.1)	3 (15.8)	3 (15.8)	5 (13.5)	0.034
GN (1 – 3)	71 (15.6)	61 (16.1)	1 (5.3)	1 (5.3)	8 (21.6)	
IgA	62 (13.6)	49 (12.9)	4 (21.1)	6 (31.6)	3 (8.1)	
Other	176 (38.7)	141 (37.1)	10 (52.6)	5 (26.3)	20 (54.1)	
Re-transplantation (%)	46 (10.1)	28 (7.4)	4 (21.1)	0 (0.0)	14 (37.8)	<0.001
DDKT (%)	230 (50.5)	192 (50.5)	10 (52.6)	9 (47.4)	19 (51.4)	0.989
Desensitization						
RTX	46 (10.1)	0 (0.0)	0 (0.0)	13 (68.4)	33 (89.2)	NA
RTX + PP	10 (2.2)	0 (0.0)	0 (0.0)	6 (31.6)	4 (10.8)	
Induction therapy						
rATG	291 (64.0)	226 (59.5)	10 (52.6)	19 (100.0)	36 (97.3)	NA
Basiliximab	164 (36.0)	154 (40.5)	9 (47.4)	0 (0.0)	1 (2.7)	
Maintenance regimen						
CsA + MMF (PD)	6 (1.3)	6 (1.6)	0 (0.0)	0 (0.0)	0 (0.0)	NA
FK + MMF (PD)	446 (98.0)	372 (97.9)	18 (94.7)	19 (100)	37 (100)	
Sirolimus/Everolimus combination	3 (0.7)	2 (0.5)	1 (5.3)	0 (0.0)	0 (0.0)	
Pre-sensitization (PRA %)						
Class I	0.0 (0.0–0.0)	0.0 (0.0–0.0)	0.0 (0.0–11.7)	37.0 (5.4–77.0)	54.0 (0.0–87.7)	<0.001
Class II	0.0 (0.0–0.0)	0.0 (0.0–0.0)	0.0 (0.0–14.0)	0.0 (0.0–63.8)	36.0 (0.0–74.3)	<0.001
HLA mismatches, median (IQR)	3.0 (2.0–4.0)	3.0 (2.0–4.0)	4.0 (2.0–4.0)	3.0 (2.2–4.8)	3.0 (2.0–4.0)	0.642

* *p* < 0.05 was considered statistically significant. IQR, interquartile range; DM, diabetes mellitus; GN, glomerulonephritis; IgA, IgA nephropathy; DDKT, deceased donor kidney transplantation; pDSA, preformed donor-specific HLA antibody; NA, not applicable; MFI, mean fluorescence intensity; RTX, rituximab; PP, plasmapheresis; rATG, recombinant anti-thymocyte globulin; CsA, cyclosporine A; MMF, Mycophenolate mofetil; PD, prednisolone; FK, FK506; PRA, panel reactive antibody.

**Table 2 jcm-09-00375-t002:** HLA classes and complement binding activities of pre- and post-transplantation donor-specific HLA antibodies in recipient groups.

HLA Classes of pDSA	Group 2: Cryptic pDSA Rebound		Group 3: pDSA Reversed		Group 4: pDSA Persistent	
	Total pDSA	C3d (+) pDSA	Total pDSA	C3d (+) pDSA	Total pDSA	C3d (+) pDSA
Pre-transplantation			19	**0 (0.0)**	37	7 (18.9)
Class I only	NA	NA	13 (68.4)	0 (0.0)	13 (35.1)	0 (0.0)
Class II* only	NA	NA	5 (26.3)	0 (0.0)	14 (37.8)	7 (18.9)
Class I + II*	NA	NA	1 (0.53)	0 (0.0)	10 (27.0)	0 (0.0)
Post-transplantation	19	4 (21.1)			37	9 (24.3)
Class I only	6 (31.6)	1 (5.3)	NA	NA	11 (29.7)	0 (0.0)
Class II* only	11 (57.9)	3 (15.8)	NA	NA	21 (56.8)	9 (24.3)
Class I + II*	2 (10.5)	0 (0.0)	NA	NA	5 (13.5)	0 (0.0)

* Class II pDSAs were assessed limited loci; as –DR, and -DQB1. Bold letters indicate significant results. pDSA, preformed donor-specific HLA antibody; MFI, mean fluorescent intensity; NA, not applicable.

**Table 3 jcm-09-00375-t003:** Comparison of the occurrence of rejection in recipient groups.

	Total	Group 1: pDSA Negative	Group 2: Cryptic pDSA Rebound			Group 3: pDSA Reversed	Group 4: pDSA Persistent			*p*-Value*
			Post-pDSA C3d (-)	Post-pDSA C3d (+)	*p*-Value		Post-pDSA C3d (-)	Post-pDSA C3d (+)	*p*-Value	
Number	455	380	15	4		19	28	9		
ACR (%)	168 (36.9)	142 (37.4)	7 (46.7)	2 (50.0)	0.906	7 (36.8)	7 (25.0)	3 (33.3)	0.624	0.479
AMR (%)	13 (2.9)	2 (0.5)	1 (6.7)	2 (50.0)	**0.035**	1 (5.3)	4 (14.3)	3 (33.3)	0.204	< **0.001**

* *p*-values among the four groups without consideration of post-DSA C3d-binding activity. *p* < 0.05 was considered statistically significant. Bold letters indicate significant results. pDSA, preformed donor-specific HLA antibody; ACR, acute cellular rejection; AMR, antibody-mediated rejection.

**Table 4 jcm-09-00375-t004:** Estimated risk of antibody-mediated rejection between recipient groups according to C3d-binding activities of post-transplant pDSAs.

Comparison groups	Odds ratio (95% CI)	95% Confidence limits		*p*-Value*	Adjusted *p*-value*
Group 1 vs. Group 2	**35.425**	5.528	227.022	**< 0.001**	**0.001**
Group 1 vs. Group 2 C3d (-)	**13.495**	1.154	157.782	**0.038**	0.076
Group 1 vs. Group 2 C3d (+)	**188.934**	17.098	>999.999	**<0.001**	**<0.001**
Group 1 vs. Group 3	10.496	0.909	121.203	0.060	0.358
Group 1 vs. Group 4	**44.084**	8.771	221.57	**<0.001**	**<0.001**
Group 1 vs. Group 4 C3d (-)	**31.489**	5.491	180.593	**<0.001**	**<0.001**
Group 1 vs. Group 4 C3d (+)	**94.467**	13.275	672.252	**<0.001**	**<0.001**
Group 2 vs. Group 3	0.296	0.028	3.142	0.313	0.999
Group 2 C3d (-) vs. Group 3	0.778	0.045	13.559	0.863	0.999
Group 2 C3d (+) vs. Group 3	**0.056**	0.003	0.923	**0.044**	0.088
Group 2 vs. Group 4	1.244	0.283	5.48	0.773	0.999
Group 2 C3d (-) vs. Group 4 C3d (-)	2.333	0.237	22.999	0.468	0.999
Group 2 C3d (-) vs. Group 4 C3d (+)	7.000	0.6	81.674	0.121	0.965
Group 2 C3d (+) vs. Group 4 C3d (-)	0.167	0.018	1.546	0.115	0.919
Group 2 C3d (+) vs. Group 4 C3d (+)	0.500	0.045	5.514	0.571	0.999
Group 3 vs. Group 4	4.200	0.477	36.978	0.196	0.999
Group 3 vs. Group 4 C3d (-)	3.000	0.308	29.182	0.344	0.688
Group 3 vs. Group 4 C3d (+)	9.000	0.781	103.723	0.078	0.156

** p*-values of < 0.05 were considered statistically significant; bold letters indicate significant results; adjusted *p*-values were computed with Bonferroni’s correction for multiple comparisons. pDSA, preformed donor-specific HLA antibody; CI, confidence interval.

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
