# Peer review of "C3d-Positive Preformed DSAs Tend to Persist and Result in a Higher Risk of AMR after Kidney Transplants"

_jcm, 2020, doi:10.3390/jcm9020375_

Round 1

Reviewer 1 Report

Just to give some examples:

Table 2 what about class II antibodies? HLA-DP antibodies were detected for (n=67). In principle this antibodies can also be DSA antibodies, but no DP typing were performed for the donors. Therefore Table 2 is not precise in counting recipients with class II antibodies. Therefore Class I should be exchange with HLA-A,B,C antibodies and class II with HLA-DR, DQ antibodies.

Is it meaningful to present extreme detailed data in the tables, for example MFI of DP antibodies. Why it is important to calculate cut-off values to predict C3d binding (or C1q binding)? What is the clinical rational or importance especially if you can measure both parameters by commercial available assay. If it is important the ROC and with corresponding AUC should be shown.

It is puzzling and sometimes difficult to compare the data which are mentioned in the text with the data of the corresponding tables: 177 recipients (38.9%) were diagnosed with rejection episodes, including ACR and AMR, with an incidence of 168 (36.9%) and 13 (2.9).  The four recipients experiencing both ACR and AMR are not shown in Table 3! In table 3, the incidence of AMR is 13 but 5 out of 15 with C3d positive antibodies experienced AMR – where are the two additional AMR.in general, it is misleading to switch between number of patients and number AMR episodes. To simplify the number of patients should be given.

.. and so on.

Thus, the whole manuscript and data presentation should be improved and written in a more focussed and precise manner.

Author Response

Just to give some examples:

Table 2 what about class II antibodies? HLA-DP antibodies were detected for (n=67). In principle this antibodies can also be DSA antibodies, but no DP typing were performed for the donors. Therefore Table 2 is not precise in counting recipients with class II antibodies. Therefore Class I should be exchange with HLA-A,B,C antibodies and class II with HLA-DR, DQ antibodies.

: We agree with your comments. In this study, only DR and DQB1 were typed in the class II loci, and thus, we added further description that the class II DSA is limited to HLA-DR and DQ loci in manuscript (line 182) and tables (table 2).

Is it meaningful to present extreme detailed data in the tables, for example MFI of DP antibodies. Why it is important to calculate cut-off values to predict C3d binding (or C1q binding)? What is the clinical rational or importance especially if you can measure both parameters by commercial available assay. If it is important the ROC and with corresponding AUC should be shown.

: Performing additional compliment binding capacity tests (such as C1q or C3d) on the SAB test for all recipients is practically difficult. For example, Koreaa national health insurance covers the C1q binding capacity test in case of suspected rejection, but in other cases, the recipients must pay 80% of the test cost. Therefore, especially in the case of preformed DSA, the prediction of compliment binding capacity through SAB MFI is important for efficient monitoring.

: As your recommendation, we added ROC and AUC as supplementary figure format (figure S1).

It is puzzling and sometimes difficult to compare the data which are mentioned in the text with the data of the corresponding tables: 177 recipients (38.9%) were diagnosed with rejection episodes, including ACR and AMR, with an incidence of 168 (36.9%) and 13 (2.9).  The four recipients experiencing both ACR and AMR are not shown in Table 3! In table 3, the incidence of AMR is 13 but 5 out of 15 with C3d positive antibodies experienced AMR – where are the two additional AMR.in general, it is misleading to switch between number of patients and number AMR episodes. To simplify the number of patients should be given.

: We deleted the sentences of low importance and confusion the readers (line 28-34).

Thus, the whole manuscript and data presentation should be improved and written in a more focussed and precise manner.

: We reaffirmed the description of this manuscript and proceeded with the overall English correction.

Reviewer 2 Report

General comment:

I have read with interest the manuscript by Choi et al. titled „Counterattack: the anamnestic response of preformed DSA and their C3d-binding capacity “. Manuscript deals with the interesting and up-to-date topic about ACR and AMR in kidney transplant individuals and focuses on identifying novel biomarkers to predict ACR/AMR development by complex study of the DSAs and their complement binding properties. The study as the whole involved 455 individuals, however, only about 1/5 of them (75 individuals) presented with DSAs at some stage of investigation. Authors then divided this group further and when combined with the presence of ACR or AMR, this led to the formation of groups containing less than 10 (sometimes even 0 or 1) patients per group. Study does not seem to me to be sized for the performed analysis and the results are hard to be generalized or interpreted. Other more specific comments are mentioned bellow. Also, the study needs to be proof-read by native English speaker as it contains many grammar errors and English tenses misuse. Some of the suggestions how to improve the English language I tried to add as comments directly into the manuscript.

Major concerns:

Title: What does the word „Counterattack “stand for? The title shall be clearer and shall contain the main observed idea of the manuscript. Current title would be more suitable for the review about the topic now for the original scientific paper Abstract shall be divided into Introduction, Methods, Results and Conclusion sections. Abstract will then gain more structure. Also, better selection of main results shall be performed for the abstract which currently contains to many incoherent data. Please, structure the abstract and choose one main message from your study for the reader. Introduction is too general and shall be more focused on the topic of the manuscript, e.g. the relation between DSA and complement and the importance of DSAs in the AMR development. Maybe summarizing figure presenting different types of DSAs and their nomenclature would be beneficial. Methodology: Line 63-64: „HLA antibody status was monitored pre- and post-transplantation at weeks 1 and 4“– this shall be better explained, it is not clear which weeks it was performed. Day 0 shall be the day of the transplantation and if samples were taken week before transplantation, it shall be explicitly stated. Current formulation is vague. It is not mentioned in the manuscript for how long patients were followed-up (from the Kaplan Mayer curves I suppose for 12 months?) Statistics: Kruskall-Walis test was used also for comparison of two groups? Why Mann-Whitney or t-test were not used? Which test were used to asses normal distribution? Why 3 different statistical software were used? Statistics in general shall be better explained. Also, power-analysis shall be included to show how many patients would indeed be needed to obtain clinically relevant data (I do realize that AMR is very very rare, however, current datasets are indeed too small). Discusion: Within the discussion section it is sometimes unclear, whether author refer to the results of the current study or whether they refer to the literature – this is partly due to inappropriate use of English tenses (i.e. mixing past tense, future tense etc.). Some suggestion how sentences may be rephrased are indicated in the manuscript, however, I advise authors to go thoroughly through the discussion and increase its inner coherence.

Minor concerns:

Please unify the use of abbreviations (all abbreviations are not introduced and some of the already introduced abbreviations are not always used which is confusing and makes manuscript hard to read) – some suggestion highlighted directly in the manuscript Introduction, line 49-50: „Using the strict definition of dnDSA, Wiebe et al. reported that no DSA developed within 6 months of transplantation “– this sentence is to general and misleading. Authors shall explain the context of Wiebe’s study. In the Methods section please indicate which method was used for eGFR estimation. CKD-EPI, Cockcroft-Gault, or another one? Unify the font size in the figure descriptions Some specific typos and English grammar mistakes are marked in the manuscript – please, review these suggestions. Table 3: Small numbers play: comparing 1 patient in one group with 2 patients in the second group with 4 patients in the third group and 3 patients in the fourth group is not of statistical significance. Table 4: Multiple comparison problematics – authors are comparing each group with each group – were the p-values adjusted for multiple comparisons? Which test was used – Kruskal-Wallis is not appropriate for this kind of comparison. Discussion: Please explain to the readers the relation between IgG subclasses and complement (e.g. how levels of IgG1/3 referred by Honger et al. correspond to your results determined by C1q-/C3d assays…) Figure S1: Why the lines between points are curved and not straight?

Author Response

General comment:

I have read with interest the manuscript by Choi et al. titled „Counterattack: the anamnestic response of preformed DSA and their C3d-binding capacity “. Manuscript deals with the interesting and up-to-date topic about ACR and AMR in kidney transplant individuals and focuses on identifying novel biomarkers to predict ACR/AMR development by complex study of the DSAs and their complement binding properties. The study as the whole involved 455 individuals, however, only about 1/5 of them (75 individuals) presented with DSAs at some stage of investigation.

Authors then divided this group further and when combined with the presence of ACR or AMR, this led to the formation of groups containing less than 10 (sometimes even 0 or 1) patients per group. Study does not seem to me to be sized for the performed analysis and the results are hard to be generalized or interpreted.

: The Authors also anticipated the sparsity problem when dividing groups further. For group comparison, we used Fisher’s exact test or Chi-square test according to the expected number of patients in each group so that the sparsity issue can be reduced. While the lack of information in some subgroups might make results hard to be generalized or interpreted, the results shown in this study strongly support the biologic function of DSA according to their characteristics and suggest that we need to pursuit further clarification of it. As you commented, the events of rejection, particularly AMR is very low although our institute is one of the most actively working on KT in Korea (over 150/year), therefore, authors believe that this retrospective data may be still helpful to understand the significance of recently introduced immune monitoring test.

2) Other more specific comments are mentioned bellow. Also, the study needs to be proof-read by native English speaker as it contains many grammar errors and English tenses misuse. Some of the suggestions how to improve the English language I tried to add as comments directly into the manuscript.

: Since this manuscript has been further revised after undergoing English correction, the authors decided to revise the native English speaker for the current version.

Major concerns:

Title: What does the word „Counterattack “stand for? The title shall be clearer and shall contain the main observed idea of the manuscript. Current title would be more suitable for the review about the topic now for the original scientific paper

: We appreciate your point about nuance of title. We changed the title of manuscript; C3d positive preformed DSAs tend to persist and have higher AMR risk after KT

Abstract shall be divided into Introduction, Methods, Results and Conclusion sections. Abstract will then gain more structure. Also, better selection of main results shall be performed for the abstract which currently contains to many incoherent data. Please, structure the abstract and choose one main message from your study for the reader.

: We reconstructed the abstract according to your advice. However, please understand that it is described as on paragraph without subheading according to JCM submission guideline.

Introduction is too general and shall be more focused on the topic of the manuscript, e.g. the relation between DSA and complement and the importance of DSAs in the AMR development. Maybe summarizing figure presenting different types of DSAs and their nomenclature would be beneficial.

: We reorganized the introduction to focus on the topic and added pictures according to your suggestions (figure 2, line 80).

Methodology: Line 63-64: „HLA antibody status was monitored pre- and post-transplantation at weeks 1 and 4“– this shall be better explained, it is not clear which weeks it was performed. Day 0 shall be the day of the transplantation and if samples were taken week before transplantation, it shall be explicitly stated. Current formulation is vague.

: We have described additional information. It, together with figure 2, will contribute to understanding the study design (line 80, line 150-155).

It is not mentioned in the manuscript for how long patients were followed-up (from the Kaplan Mayer curves I suppose for 12 months?)

: We added information about follow-up period to assess the clinical outcome (, line 150-155).

Statistics: Kruskall-Walis test was used also for comparison of two groups? Why Mann-Whitney or t-test were not used? Which test were used to asses normal distribution?

: For continuous variables in patient characteristics (Table 1), four-group comparison was conducted with Kruskal-Wallis test or one-way ANOVA. For continuous variables in donor-specific antibody characteristics (Table S1), two-subgroup comparison was performed within each of Group 2 and Group 4, using Mann-Whitney U-test in that the sample sizes of subgroups were very small.

Why 3 different statistical software were used? Statistics in general shall be better explained.

: Most of the analyses was performed using SAS and SPSS based on availability, and some descriptive graphic analysis was done with Analyse-it software because SAS and SPSS do not support the user friendly graphic presentation.

The Authors added more details of statistical analyses in the Methods, including the above responses to the Reviewers (line 129-135, 167-169).

Also, power-analysis shall be included to show how many patients would indeed be needed to obtain clinically relevant data (I do realize that AMR is very very rare, however, current datasets are indeed too small).

: As you commented, the events of rejection, particularly AMR is very low although our institute is one of the most actively working on KT in Korea (over 150/year). And the positive predictive value of complement binding assay has not been established yet and different publications are reporting variable data. It might be the reason that it is hard to find the prospective controlled trial for investigation of the clinical significance of HLA antibodies including complement binding ones.

Discusion: Within the discussion section it is sometimes unclear, whether author refer to the results of the current study or whether they refer to the literature – this is partly due to inappropriate use of English tenses (i.e. mixing past tense, future tense etc.). Some suggestion how sentences may be rephrased are indicated in the manuscript, however, I advise authors to go thoroughly through the discussion and increase its inner coherence.

: the authors decided to revise the native English speaker for the current version.

Minor concerns:

Please unify the use of abbreviations (all abbreviations are not introduced and some of the already introduced abbreviations are not always used which is confusing and makes manuscript hard to read) – some suggestion highlighted directly in the manuscript Introduction, line 49-50: „

: We checked overall abbreviations.

Using the strict definition of dnDSA, Wiebe et al. reported that no DSA developed within 6 months of transplantation “– this sentence is to general and misleading. Authors shall explain the context of Wiebe’s study.

: We added explain the context of Wiebe’s study (line 54-59).

In the Methods section please indicate which method was used for eGFR estimation. CKD-EPI, Cockcroft-Gault, or another one?

: eGFR was calculated using the modification of diet in renal disease (MDRD) study equation (line 163).

Unify the font size in the figure descriptions Some specific typos and English grammar mistakes are marked in the manuscript – please, review these suggestions.

: We checked font size and the authors decided to revise the native English speaker for the current version.

Table 3: Small numbers play: comparing 1 patient in one group with 2 patients in the second group with 4 patients in the third group and 3 patients in the fourth group is not of statistical significance.

: It is very difficult to obtain enough targets to assess the clinical outcome of preformed DSA because recent immunosuppression protocol and monitoring is very effective. The Authors also anticipated the sparsity problem when dividing groups further. For group comparison, we used Fisher’s exact test or Chi-square test according to the expected number of patients in each group so that the sparsity issue can be reduced. While the lack of information in some subgroups might make results hard to be generalized or interpreted, it confirmed that there are significant differences between groups when analyzed using several statistical methods.

Table 4: Multiple comparison problematics – authors are comparing each group with each group – were the p-values adjusted for multiple comparisons?

: The Author used Bonferroni’s correction to adjust multiple comparison issue, and add the related information into the Methods section and the adjusted p-values in Table 4 as well.

Which test was used – Kruskal-Wallis is not appropriate for this kind of comparison.

: The Author conducted the univariable logistic regression repeatedly for four-group comparison (Groups 1, 2, 3 & 4) and also for two- or four-subgroup comparisons (e.g., Group 1, Group 4 C3d(-) & Group 4 C3d(+)). Wald’s chi-square test was used for pairwise comparison with Bonferroni’s correction.

Discussion: Please explain to the readers the relation between IgG subclasses and complement (e.g. how levels of IgG1/3 referred by Honger et al. correspond to your results determined by C1q-/C3d assays…)

: We tried to add more description to these issues (line 294-312).

Figure S1: Why the lines between points are curved and not straight?

: The spline interpolation method was used to construct the curved graph, which is known to have less interpolation error than the polynominal interpolation method.

Round 2

Reviewer 1 Report

No further comments.

Reviewer 2 Report

I have read with interest the revised version of the manuscript. Authors definitelly improved the manuscript quality, did a great job and answered all my querries. I have just one minor comment, that shall be adressed, otherwise, there are no further comments.

Minor concerns:

Title is clearer now, however, please, do not use abbreviations of the title (KT, DSA)

Thanks to the authors for the great improvement of the manuscript!